# Current Trends in Stroke Biomarkers: The Prognostic Role of S100 Calcium-Binding Protein B and Glial Fibrillary Acidic Protein

**DOI:** 10.3390/life14101247

**Published:** 2024-10-01

**Authors:** Georgios Anogianakis, Stylianos Daios, Nikolaos Topouzis, Konstantinos Barmpagiannos, Georgia Kaiafa, Athena Myrou, Eleftheria Ztriva, Alexandra Tsankof, Eleni Karlafti, Antonia Anogeianaki, Nikolaos Kakaletsis, Christos Savopoulos

**Affiliations:** 1First Propaedeutic Department of Internal Medicine, AHEPA University General Hospital of Thessaloniki, Aristotle University of Thessaloniki, 54636 Thessaloniki, Greece; anogian@auth.gr (G.A.); stylianoschrys.daios@gmail.com (S.D.); topouzisnikolaos@gmail.com (N.T.); kostasmparmp@hotmail.gr (K.B.); gdkaiafa@yahoo.gr (G.K.); taniamyrou@gmail.com (A.M.); elztriva@gmail.com (E.Z.); a.tsankof@gmail.com (A.T.); linakarlafti@hotmail.com (E.K.); aanogeianaki@gmail.com (A.A.); kakaletsisnikos@yahoo.gr (N.K.); 2Department of Emergency, AHEPA University General Hospital of Thessaloniki, Aristotle University of Thessaloniki, 54636 Thessaloniki, Greece

**Keywords:** stroke, biomarker, S100B, GFAP

## Abstract

Stroke is the third leading cause of death in the developed world and a major cause of chronic disability, especially among the elderly population. The major biomarkers of stroke which are the most promising for predicting onset time and independently differentiating ischemic from hemorrhagic and other stroke subtypes are at present limited to a few. This review aims to emphasize on the prognostic role of S100 calcium-binding protein b (S100B), and Glial Fibrillary Acidic Protein (GFAP) in patients with stroke. An electronic search of the published research from January 2000 to February 2024 was conducted using the MEDLINE, Scopus, and Cochrane databases. The implementation of S100B and GFAP in existing clinical scales and imaging modalities may be used to improve diagnostic accuracy and realize the potential of blood biomarkers in clinical practice. The reviewed studies highlight the potential of S100B and GFAP as significant biomarkers in the prognosis and diagnosis of patients with stroke and their ability of predicting long-term neurological deficits. They demonstrate high sensitivity and specificity in differentiating between ischemic and hemorrhagic stroke and they correlate well with stroke severity and outcomes. Several studies also emphasize on the early elevation of these biomarkers post-stroke onset, underscoring their value in early diagnosis and risk stratification. The ongoing research in this field should aim at improving patient outcomes and reducing stroke-related morbidity and mortality by developing a reliable, non-invasive diagnostic tool that can be easily implemented in several healthcare settings, with the ultimate goal of improving stroke management.

## 1. Introduction

Stroke is the third leading cause of death in the developed world after coronary artery disease and all types of cancer. It is also a major cause of chronic disability, especially among the elderly population. It is estimated that it caused close to 6.5 million deaths worldwide in 2015, a number that is projected to reach approximately 8 million by 2030 [1]. Regarding the epidemiology of stroke, it must be noted that the incidence of stroke has been studied over the years in many countries by means of stroke registries. The annual stroke rate has appeared in different studies in the range of 1.35–4 per 1000. It has been shown that 70% of strokes are due to cerebral ischemia, 27% to cerebral hemorrhage and 3% to unknown causes. Only 10% are attributable to carotid atherosclerotic disease. Most stroke survivors experience permanent neurological damage, significantly impacting their ability to work and overall quality of life [2].

The clinical course of the disruption of oxygen supply to the brain was first charted in 1943, when it was demonstrated that deprivation of oxygen influences the normal function of the neocortex with remarkable speed [3]. The time required to achieve a marked disruption of function is only a few seconds, i.e., much shorter than the minutes needed before the first histopathological evidence of neuronal damage is evident [4,5]. Therefore, it is apparent that the earliest effects of oxygen deprivation entail a rapid influence on the excitability and synaptic mechanisms that are the basis of cortical function [6,7].

Because of the characteristic histological structure of each gross anatomical part of the brain there is a differential response of the different neuronal populations to stressful neurodegenerative conditions, a phenomenon called selective neuronal vulnerability (SNV). SNV refers to the differential sensitivity of neuronal populations in the CNS to stresses that cause cell injury or death and lead to neurodegeneration. Neurons, e.g., in the entorhinal cortex, in the CA1 region of the hippocampus, in the frontal cortex, and in the amygdala are most sensitive to the neurodegeneration due to acute or chronic ischemic and/or oxidative stress [8,9,10,11].

In this respect it should be noted that the intense need for oxygen-consumption makes the mammalian brain highly vulnerable to hypoxia [12]. In general, hypoxia impairs several cognitive domains such as attention, learning and memory, processing speed and executive function. The mechanism is similar in both acute and chronic hypoxia and it includes effects of oxidative stress, such as calcium overload [13] (whereby altered calcium homeostasis induces cell damage), adenosine [14] (whose release attempts to mitigate oxidative stress), mitochondrial disruption (as mitochondria are the primary targets of hypoxic injury, and hypoxia pathology originates from the initial mitochondrial dysfunction) [15], inflammation, and excitotoxicity.

## 2. Biomarkers

Use of the term “biomarker” describes molecules that can be used to measure and predict the course of biological function [16]. According to the US National Institutes of Health a biomarker is “a characteristic that is objectively measured and evaluated as an indicator of normal biological processes, pathogenic processes, or pharmacologic responses to a therapeutic intervention” [17]. However, although the term “biomarker” can refer to clinical or imaging data, it usually describes molecules found in blood.

Genome-wide informatic analysis identified several proteins (e.g., Glial Fibrillary Acidic Protein (GFAP), Myelin basic protein (MBP), b-synuclein, OPALIN, Metallothionein isoform 3 (MT-3), Synaptosome-associated protein 25, Kinesin family member 5A, Myelin-associated oligodendrocyte basic protein as potential blood biomarkers of neurological injury. In addition, four of these proteins (MT-3, SNAP-25, KIF5A, b-synuclein) and several mRNAs and miRNAs show strong associations with infarct volume [18]. However, the major biomarkers for stroke which have been presently identified and are most promising for independently differentiating ischemic stroke from hemorrhage and mimics (i.e., distinguishing between stroke subtypes), identifying large vessel occlusion, and predicting stroke onset time are at present limited to S100 calcium-binding protein b (S100B), which is a calcium-binding protein that is thought to indicate brain tissue damage [19]; GFAP, which is an abundant glial structural protein that is thought to indicate brain tissue damage; Nucleosomes, which have been found to correlate significantly with clinical status at admission [20,21] and MBP; elevated levels of MBP have been reported in 39% of the patients upon admission [22].

Biological markers have been extensively studied since 1977 when the term ‘biological marker’ first appeared in MEDLINE. However, most molecular markers of hypoxia related to stroke are neuronal markers for degenerative diseases of the nervous system rather than stroke-specific markers.

Despite the advances in imaging and diagnostic procedures, the accurate prediction of the onset of stroke and the differentiation between the ischemic and the hemorrhagic subtypes remains challenging. Current biomarkers are promising but are limited in their ability to provide timely and precise information.

The aim of the present review, therefore, is to analyze and summarize the recent relevant clinical literature on molecular markers of focal hypoxia with particular emphasis on the prognostic role of S100B protein and the GFAP in patients with stroke.

## 3. Methods

### 3.1. Search Strategy

A systematic electronic search of the published research from January 2000 to February 2024 was conducted using the MEDLINE, Scopus, and Cochrane databases. The Medical Subject Headings and keywords used as search terms were: (“proteomics” OR “S100 calcium-binding protein b’’ OR “S100B’’ OR “S100” OR “Glial fibrillary acidic protein” OR “GFAP”) and (“stroke’’ OR “ischemic stroke”) and (‘’outcome’’ OR ‘’prognosis’’ OR “prognostic value” OR “NIH Stroke Scale” OR “NIHSS” OR “Barthel” OR “Barthel index” OR “infarct” or “infarct volume” OR “volume” OR “recurrence” OR “mortality”). The reference lists of the included studies and relevant reviews were also hand-searched to identify further relevant studies. This review was conducted in accordance with ethical guidelines and standards. No new data involving human or animal subjects were generated in this review. All data discussed in this review are available from the cited sources. No new data were generated for this study.

### 3.2. Study Selection—Eligibility Criteria

The eligibility of the retrieved studies was independently assessed by two investigators (N.T., K.B.) according to prespecified criteria. The most relevant full-text articles investigating the prognostic significance of S100B and GFAP (measured either as dichotomous or continuous variables) in patients with ischemic stroke, irrespective of the stroke type, were included in this review. Abstracts without complete published papers, case reports, editorials, and letters were excluded. Any discrepancies were resolved by consensus or by the involvement of a third reviewer (S.D). All publications, most of them within the last 5 years were selected based on completeness, relevance, and new contributions to the question of distinguishing between stroke subtypes, while also showing promise for predicting thrombolysis and thrombectomy outcome in patients with acute ischemic stroke (AIS).

## 4. The Contribution of S100B (Table 1) and GFAP (Table 2)

### 4.1. S100B Contribution

S100B is a homodimeric protein that is primarily found in brain extracts, consisting of two beta subunits, each weighing between 9–14 kDa (Figure 1). Extensive research has elucidated its distribution in astrocytes and various other glial cell types, including oligodendrocytes, Schwann cells, ependymal cells, retinal Muller cells, and enteric glial cells. In addition, it has been detected in specific neuron subgroups, indicating a broader distribution beyond the nervous system [19].

**Figure 1 life-14-01247-f001:**
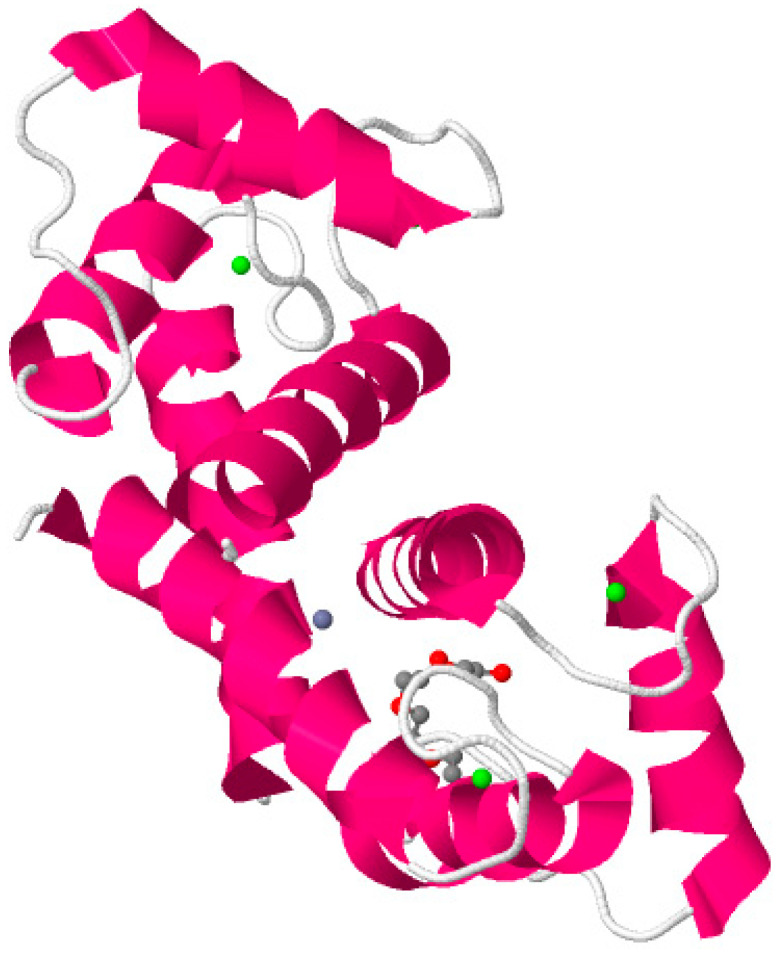
3D Structure of S100B. 3D structure of S100B was obtained from the RCSB PDB (RCSB.org) of PDB ID 3D0Y [23].

**Table 1 life-14-01247-t001:** The role of S100B.

No.	Author/Ref. Number	Title	Year	N	Study Design	Stroke Subtypes	Results
1	Foerch et al. [24]	Elevated serum S100B levels indicate a higher risk of hemorrhagic transformation after thrombolytic therapy in acute stroke	2007	275	Retrospective study	IS	-Serum S100B values were significantly higher in patients with hemorrhagic transformation compared to patients without
2	J. Montaner et al. [25]	A panel of biomarkers including caspase-3 and D-dimer may differentiate acute stroke from stroke-mimicking conditions in the emergency department	2010	915 strokes and 90 mimics	Single-center, prospective observational study	IS and ICH	-S100B play a role for distinguishing ischaemic from haemorrhagic stroke
3	Knauer et al. [26]	A biochemical marker panel in MRI-proven hyperacute ischemic stroke-a prospective study	2012	174	Prospective cohort study	IS	-Approximately 98% of the patients did not reach the lower limit of the testing range for S100B
4	Selçuk et al. [27]	The Relationship of Serum S100B Levels with Infarction Size and Clinical Outcome in Acute Ischemic Stroke Patients	2014	50 patients, 26 controls	Prospective, case-control study	IS	-S100B levels showed significantly higher values than the control group -Alteration of S100B levels did not show any significant differences between the 1st to 3rd days and the 1st to 5th days, but were significantly higher on the 3rd day compared to the 5th day -S100B levels were correlated with infarct volume, especially on the 3rd day -Weak correlation between the first month mRS score and S100B levels of the 3rd day -no significant relationship between the concurrent NIHS scores and S100B levels
5	Deboevere et al. [28]	Value of copeptin and the S-100b protein assay in ruling out the diagnosis of stroke-induced dizziness pattern in emergency departments	2019	135	Single-center, prospective, observational study	N/A	-S100 levels above normal values were more frequent in patients with stroke than in those without. -Absence of S100 elevation seems to rule out the diagnosis of stroke
6	Garzelli et al. [29]	Secondary S100B Protein Increase Following Brain Arteriovenous Malformation Rupture is Associated with Cerebral Infarction	2020	216 patients	Single-center, retrospective study	IS	-Secondary S100B protein serum elevation was found in 17.1% of ruptures and was associated with secondary infarction, vasospasm-related infarction, intensive care, and hospital length of stay, but not with early re-bleeding or in-hospital mortality
7	Iwamoto et al. [30]	Predicting hemorrhagic transformation after large vessel occlusion stroke in the era of mechanical thrombectomy	2021	91	Single-center, prospective, observational study	IS	-S100B levels were not associated with the development of relevant hemorrhagic transformation, with neurological deterioration/functional outcomes, or with parenchymal hematoma

GFAP, Glial Fibrillary Acidic Protein; IS, ischemic stroke; ICH, intracerebral hemorrhage; S100B, S100 calcium-binding protein B; UCH-L1, Ubiquitin C-terminal hydrolase L1.

In the late 1970s, the presence of S100B protein in the extracellular space was initially observed when elevated levels were identified in the cerebrospinal fluid (CSF) of patients in the acute phase of multiple sclerosis, in contrast with lower levels that were detected during the disease’s stable phase. This indicated the evaluation of S100B levels in bodily fluids as a potential biomarker for nervous system cell damage (Figure 2). Subsequently, research on S100B as an indicator of brain injury has expanded to include other bodily fluids beyond CSF [19].

The difficulty of differentiating ischemic stroke in the hyperacute phase has led to the exploration of several blood-based biomarkers. The present state of research indicates that the proteins MT-3, SNAP-25, KIF5A, b-synuclein are associated with infarct volume [36]. Blood sampling for S100, GFAP, NR2, IL6, and BNP can potentially differentiate intracerebral hemorrhage (ICH) and IS, but their overall discriminatory ability is low [37]. Measurement of S100B on day 3 after acute cerebrovascular stroke is significantly correlated with short-term functional outcome on day 14. It increases post-stroke but has low specificity for AIS due to its tendency to be raised from extracranial sources. Its levels peak 3 days after symptom onset and correlate well with infarct volume and functional outcome [38]. It can also predict a malignant course of infarction in the proximal occlusion of the medial cerebral artery [31].

While S100B is not considered a valuable biomarker for diagnosing AIS, it may serve a more useful role as an additional tool for identifying patients at increased risk of specific early neurological complications after a stroke. It could act as a surrogate marker of infarct size and functional outcome [27]. More specific, serum S100B levels measured within 24 h of symptom onset showed an independent correlation with the occurrence of symptomatic intracranial hemorrhage and brain edema in patients with AIS [24]. In addition, increased concentration of S100B before thrombolytic therapy have been found to be an independent risk factor for hemorrhagic transformation in stroke patients. Despite that, the diagnostic accuracy of S100B is too low for it to serve as a reliable biomarker in clinical practice [24]. In patients with ruptured brain arteriovenous malformations, an early elevation in S100B protein serum levels is associated with a poor prognosis. A secondary elevation of S100B protein serum levels has been linked to secondary infarction in these patients [29].

S100B levels assessed two days after mechanical thrombectomy for AIS has been also proven valuable in differentiating between favorable and unfavorable functional outcomes [39] Luger et al. observed that successful recanalization led to low S100B levels in individuals that intervention prevented a final infarct, while those who still developed infarcts despite recanalization exhibited high S100B levels [39]. In a similar manner to traumatic brain injury and subarachnoid hemorrhage, functional recovery in ischemic stroke patients can be anticipated by assessing serum S100B. More specific, concentrations of S100B between days 2 and 4 following acute stroke onset may serve as predictors for both neurological status and functional impairment at discharge [32].

S100B protein also constitutes a reliable biomarker of neural distress and a Damage-Associated Molecular Pattern (DAMP) molecule that triggers tissue reaction to damage in various neural disorders. The protein levels and/or distribution are related to the progress of different neural disorders, such as acute brain injury, neurodegenerative diseases, congenital/perinatal disorders, psychiatric disorders, and inflammatory bowel disease. Therefore, S100B protein could be a potential therapeutic target for these neural disorders, as its overexpression/administration worsened the disease, while its deletion/inactivation improved it [33].

### 4.2. GFAP Contribution

GFAP is a structural protein in mature astrocytes in the central nervous system, characterized by a filament length of approximately 8–9 nm. The gene encoding GFAP is located on chromosome 17q21. In normal astrocytes, GFAP is expressed as a non-soluble monomeric protein comprising 432 amino acids with a molecular weight of 49.8–53 kDa (Figure 3). Due to its exclusive production by astrocytes, GFAP is uniquely located in the brain and demonstrates prognostic role in patients with stroke (Figure 4) [40,41].

**Figure 3 life-14-01247-f003:**
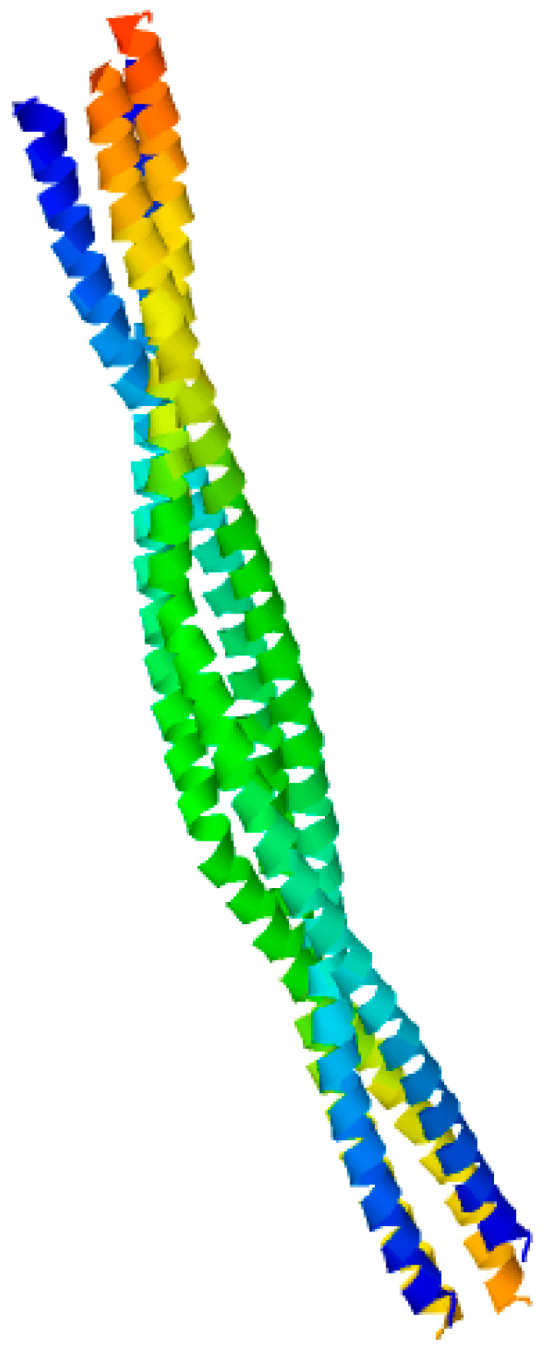
3D Structure of GFAP. 3D structure of GFAP was obtained from the RCSB PDB (RCSB.org) of PDB ID 6A9P [42].

**Figure 4 life-14-01247-f004:**
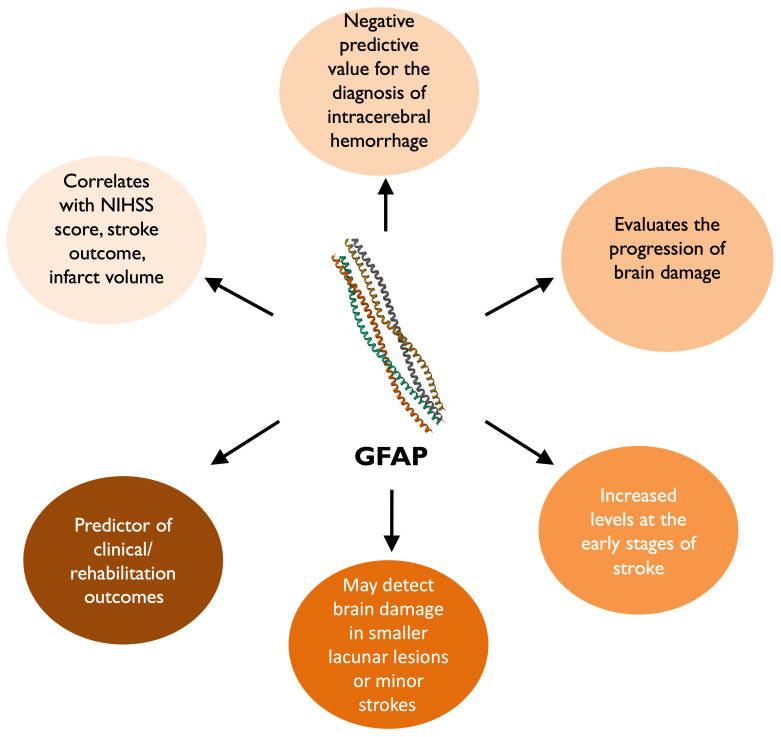
The role of GFAP in patients with stroke [37,43,44,45,46,47].

**Table 2 life-14-01247-t002:** The role of GFAP.

No.	Author/Ref. Number	Title (GFAP)	Year	N	Study Design	Stroke Subtypes	Results
1	Herrmann et al. [41]	Release of glial tissue-specific proteins after acute stroke: A comparative analysis of serum concentrations of protein S-100B and glial fibrillary acidic protein	2000	32	Cross-sectional study	IS	-GFAP was associated with the volume of brain lesions and the neurological status at discharge from the hospital -GFAP was found to be a more sensitive marker of brain damage in patients with smaller lacunar lesions or minor strokes
2	Ehrenreich et al. [46]	Circulating damage marker profiles support a neuroprotective effect of erythropoietin in ischemic stroke patients	2011	163	Randomized Controlled Trial	IS	-EPO-treated patients had significantly lower concentrations over 7 days of observation, as reflected by the composite score of all three markers and by UCH-L1. S100B and GFAP showed a similar tendency.
3	Bhatia et al. [37]	Role of Blood Biomarkers in Differentiating Ischemic Stroke and Intracerebral Hemorrhage	2020	250	Cross-sectional	187 IS, 63 ICH	-GFAP showed low overall discriminatory ability with an AUC of 56%
4	Misra et al. [48]	Blood biomarkers for the diagnosis and differentiation of stroke: A systematic review and meta-analysis	2020	8085	Meta-analysis	5001 IS, 756 ICH	-GFAP successfully differentiated ischemic stroke from intracerebral hemorrhage within 6 h.
5	O’Connell et al. [36]	Newly identified blood biomarkers of neurological damage are correlated with infarct volume in patients with acute ischemic stroke	2021	43	Cross-sectional	IS	-Correlation of GFAP with infarct volume
6	Correia et al. [49]	Early plasma biomarker dynamic profiles are associated with acute ischemic stroke outcomes	2022	54	Cross-sectional	IS	-GFAP levels exhibited an early and prominent increase between admission and just after treatment. -GFAP rate-of-change between admission and immediately after recanalization had a good discriminative capacity between clinical outcomes. -In patients with admission CT-ASPECTS <9, GFAP rate-of-change was good outcome predictor
7	Sayad et al. [21]	A magnetoimpedance biosensor microfluidic platform for detection of glial fibrillary acidic protein in blood for acute stroke classification	2022	52	Technical paper (method for GFAP detection in biofluids)	-	-Detection of recombinant GFAP protein in phosphate-buffered saline and in mouse blood samples (detection limit 0.01 ng/mL) and of physiological GFAP in blood and plasma samples (detection limit 1.0 ng/mL) obtained from acute stroke patients
8	Pujol-Calderón et al. [43]	Prediction of Outcome After Endovascular Embolectomy in Anterior Circulation Stroke Using Biomarkers	2022	90	Longitudinal observational study	IS	-At 3 months, GFAP levels were back to normal. -GFAP correlated well to outcome, as well as to infarct volume and NIHSS at 24 h. -The combination of NIHSS at 24 h with either tau, NFL or GFAP at 48 h gave the best poor outcome prediction.
9	Ferrari et al. [47]	Quantification and prospective evaluation of serum NfL and GFAP as blood-derived biomarkers of outcome in acute ischemic stroke patients	2023	36	Longitudinal observational study	IS	-GFAP showed an earlier peak on first day after stroke -GFAP correlated with clinical/rehabilitation outcomes both longitudinally and prospectively. -Multivariate analysis revealed that GFAP on the first day was an independent predictor of 3-month NIHSS, TCT, FAC and FIM scores
10	Florijn et al. [50]	Non-coding RNAs versus protein biomarkers to diagnose and differentiate acute stroke: Systematic review and meta-analysis	2023	20,678	Meta-analysis	11,627 IS, 2110 ICH	-Circulating microRNAs and proteins perform equally well in the diagnosis of ischemic stroke -GFAP differentiated subtypes of stroke -A biomarker panel of GFAP and UCH-L1 improved the sensitivity and specificity of UCH-L1 alone to differentiate stroke.

GFAP, Glial Fibrillary Acidic Protein; IS, ischemic stroke; EPO, erythropoietin; S100B, S100 calcium-binding protein B; UCH-L1, Ubiquitin C-terminal hydrolase L1.

In studies focusing on GFAP, lesion topography and infarct volume are typically evaluated by CT scans, while clinical status and neurological outcome are assessed using NIHSS and the Barthel Index [41]. These studies often involve a comparative analyses of serum concentrations of GFAP and S100B in patients with AIS, particularly in those with cerebral ischemia in the anterior territory of vascular supply. Blood samples are usually collected pre-treatment and at intervals post-endovascular treatment [43]. Although most studies have sought to determine differences between AIS and ICH, certain studies have primarily focused on time-dependent changes of GFAP in patients with AIS [49,51]. These studies revealed that the transient discharge of astroglial proteins, including GFAP, into the CSF, might signify localized ischemic injury and subsequent astroglial cell degeneration in the penumbra region [52]. Notably, only a study by Hu et al. compared GFAP serum levels in AIS patients with healthy controls and determined increased levels at the early stage of stroke with a certain correlation to the severity of cerebral infarction [44]. Nonetheless, their severity stratification into three groups only aligned with occlusion outcomes, disregarding differentiation based on the underlying cause of vessel occlusion.

GFAP levels are found to be positively correlated with stroke outcome, infarct volume, and NIHSS scores in patients with AIS [43]. Both GFAP and S-100B show time-dependent increases post-stroke. Combining NIHSS at 24 h with GFAP at 48 h yields the best prediction for negative outcomes. Several studies also showed a positive correlation between GFAP and NIHSS score. They highlighted that the higher the value of GFAP serum levels, the higher the value of NIHSS score [44,45]. Early increases in GFAP levels upon hospital admission are observed, with higher negative predictive value for diagnosing ICH [37].

GFAP is associated with the volume of brain lesions and neurological status at discharge. The strongest correlation between biomarker serum concentrations and Barthel score occurs at 4 days post-stroke onset [45]. GFAP demonstrates higher sensitivity in detecting brain damage in smaller lacunar lesions or minor strokes [45]. Stroke subtypes can be differentiated using both biomarkers, and a biomarker panel including GFAP and UCH-L1 enhances sensitivity and specificity [50].

In endovascular treatment, GFAP and S100B can be used to evaluate the progression of brain damage and their relationship to outcome. In this respect, patients from the Göttingen EPO Stroke Study that did not receive thrombolysis but were treated with erythropoietin (EPO) showed significantly lower GFAP concentrations over 7 days, suggesting a beneficial effect of EPO in patients with IS [46]. At 3 months, GFAP levels are back to normal. Also, GFAP levels correlate with various clinical/rehabilitation outcomes, making it an independent predictor of 3-month NIHSS, TCT, FAC, and FIM scores [47].

## 5. Discussion

### 5.1. Single Biomarkers and Criteria of Patient Progression

The preceding discussion underlines the fact that biomarkers hold significant promise in elucidating the pathophysiology of various neurological disorders, charting their natural history, predicting the outcome of acute cerebrovascular incidents, and potentially informing therapeutic interventions. However, even though biomarkers have emerged as valuable tools in disease diagnosis, prognosis, and evaluation of the treatment response across various medical conditions, in the context of ischemic stroke, their clinical utility remains uncertain, primarily due to limitations in sensitivity, specificity, and the lack of validated predictive models.

The criteria that different papers used to measure efficiency and efficacy of biomarkers in assessing patient progression after IS or ICH are tabulated in Table 3. Most studies used the NIHSS criteria, although the ASPECTS, the Barthel Index score, the CoRisk score, the HINTS tests, the modified Rankin score (mRS), the NIHSS, the OSCP, the Prehospital Stroke Score (PreSS), and the TOAST criteria were also used, based on the particular study setting.

The integration of S100B and GFAP with existing clinical scales can improve the accuracy and predictive power in assessing stroke outcomes. They can also contribute to the early identification of high-risk patients, allowing for timely interventions. For instance, patients with increased NIHSS score and elevated biomarker levels might be prioritized for intensive monitoring and more aggressive therapeutic management. In addition, by incorporating these biomarkers into the established clinical scales, a more nuanced understanding of stroke severity and prognosis might be achieved, ultimately leading to improved patient outcomes.

The ideal stroke biomarker should possess several key characteristics, including high sensitivity, high likelihood ratio positive, high diagnostic odds ratio, and early detection in blood. Such a biomarker would enable differentiation between ischemic and hemorrhagic stroke, thus guiding appropriate treatment strategies. Additionally, a prognostic biomarker capable of predicting hemorrhagic transformation risk would aid in optimizing therapeutic interventions and improving patient outcomes.

### 5.2. The Role of S100B

S100B has gathered considerable attention due to its multifaceted roles in neural injury and inflammation [33]. The reason for this is that despite the heterogeneity of the diverse etiologies and clinical manifestations of neurological diseases, the involvement of inflammatory pathways, characterized by the expression and secretion of pro-inflammatory molecules, is a recurrent theme across various neurological disorders. S100B, presents a DAMP, shares characteristics with inflammatory molecules and apparently modulates immune responses and inflammatory cascade [33].

Genetic factors can also be involved in the development of IS. To date, only one study by Lu et al. demonstrated that S100B gene rs9722 polymorphism may increase the risk of IS in Chinese population, most likely by enhancing the expression of serum S100B. Several other polymorphisms that were evaluated in this study failed to show an association. However, further studies with larger sample sizes across various ethnic groups are required to confirm and empower the findings of this study [62].

#### 5.2.1. S100B Role in Inflammation and IS/ICH Differentiation

The pro-inflammatory properties include the induction of migration and activation of microglial cells, and the promotion of a pro-inflammatory phenotype in astrocytes [63]. The upregulation of inflammatory mediators, such as cytokines and chemokines, in response to S100B stimulation underscores S100B’s role in driving neuroinflammation [33,34]. Studies have reported promising sensitivity and specificity for S100B in distinguishing between IS and ICH, with notable accuracy in predicting short-term functional outcomes post-stroke. However, the widespread elevation of S100B in other neurological and neuropsychological disorders, such as Alzheimer’s disease and schizophrenia, in addition to the fact that elevated levels of S100B can arise from extracranial sources, raises concerns about its diagnostic accuracy and its specificity for stroke triage [33].

Despite this drawback, longitudinal studies have provided valuable insights into the kinetics of serum S100B levels following AIS. Contrary to expectations, serum S100B levels do not rise immediately after AIS onset but peak around 3 days post-symptom onset [38]. Notably, these elevated levels correlate well with infarct volume and are higher in stroke patients at risk of malignant infarction or hemorrhagic transformation following thrombolysis [31].

#### 5.2.2. The Prognostic Role of S100B

While S100B may not serve as a reliable biomarker for diagnosing AIS, its potential utility lies in identifying patients at increased risk of early neurological complications post-stroke and predicting functional outcomes. Particularly in non-specialist hospitals, S100B could then serve as an additional tool to guide clinical decision-making and stratify patients based on their risk profile. Thus, while S100B may not meet the criteria of a clinically informative biomarker for AIS diagnosis, its role in predicting early neurological complications and functional outcomes is promising [35].

One should always keep in mind that the complexity of stroke pathophysiology complicates the identification of a single biomarker that fulfills all the criteria that were previously mentioned as the hallmarks of the ideal stroke biomarker. Existing biomarkers exhibit varying degrees of diagnostic accuracy and prognostic value, but no one possesses comprehensive capabilities across all aspects of stroke management. For instance, biomarkers like ATIII, fibrinogen, and IMA index demonstrate high diagnostic accuracy in early stroke detection but lack the ability to differentiate between ischemic and hemorrhagic stroke or predict hemorrhagic transformation risk [64].

### 5.3. GFAP Role in Differentiation between Stroke Subtypes and Differentiation

Conversely, biomarkers like GFAP show promise in distinguishing stroke subtypes, predicting hemorrhagic transformation and aids in distinguishing between hemorrhagic and ischemic strokes based on the extent of blood-brain barrier disruption [51]. This differentiation is crucial for the acute management of patients with stroke, since acute therapeutic options like thrombolysis are suitable for IS, but they are contraindicated in ICH. Identifying the stroke subtype in a rapid manner with the contribution of GFAP could therefore expedite appropriate treatment, potentially reducing mortality and improving functional outcomes. Similar considerations hold when examining biomarkers as the window into stroke pathophysiology. For example, changes in ATIII and fibrinogen levels signify endothelial damage following vessel thrombosis or rupture, offering insights into stroke etiology. Finally, matrix metalloproteinase-9 (MMP-9) and S100B provide valuable information about the state of the blood-brain barrier and the likelihood of hemorrhagic transformation, informing risk stratification and treatment decisions [48,65].

#### The Prognostic Role of GFAP and Neurological Status

GFAP seems to demonstrate also a prognostic role, as it is associated with the extent of brain damage and neurological status at discharge [45]. Surjawan showed in an observational prospective study of 74 participants that GFAP measured at 48 to 72 h after the onset of the episode was significantly associated with NIHSS at discharge [45]. Ferrari et al. also highlighted the value of GFAP as a predictor of several scores used in patients with IS such as NIHSS, TCT, FAC, and FIM in a prospective cohort of 36 patients within a 3 month follow-up period [47]. This suggests that GFAP can be a useful biomarker for predicting long-term disability contributing to patient risk stratification and eventually leading to a tailored post-stroke care.

### 5.4. Limitations

Although the results of the studies are promising, the clinical application of these biomarkers is characterized by several limitations. A notable variability in the sensitivity and specificity along the conducted studies is evident and there is an obvious lack of standardized measurement protocols. Several studies are investigating the application of handheld, point-of-care devices in emergency setting that can rapidly measure and provide results in less than 15 min, but in most cases, a central laboratory is mandatory. In addition, the specific cost varies depending on the method used. Therefore, more well-designed large-scale studies in diverse patient populations are required before these biomarkers can be used as a standalone diagnostic tool.

### 5.5. Biomarker Panels and Future Directions

As the role of blood-based biomarkers in predicting outcomes in IS became clearer, it resulted in a relative surge in publications investigating their prognostic value. However, despite the surge in publications, their critical analysis reveals persistent shortcomings in their methodological and statistical quality, i.e., the main element hindering their translation into clinical practice.

While the quest for a single biomarker capable of meeting all the diagnostic and prognostic needs of stroke remains elusive, it appears that by leveraging the complementary capabilities of multiple biomarkers, clinicians may achieve more accurate and comprehensive stroke diagnosis, helping to improve patient outcomes. Thus, studies have demonstrated that a panel of biomarkers, including complement C3, high-sensitive C-reactive protein, hepatocyte growth factor, MMP9, and anti-phosphatidylserine antibodies, can provide more comprehensive risk stratification for adverse outcomes in ischemic stroke [66].

In addition, the integration of blood-based biomarkers into predictive models can potentially refine clinical decision-making in IS management. Present predictive models are primarily based on demographic and clinical parameters. This means that they are susceptible to bias and lack the accuracy required for reliable clinical decision-making. In this respect, blood-based biomarkers and biomarker panels offer the opportunity to supplement these models with additional prognostic information, enhancing risk stratification and informing tailored management strategies for IS patients.

Finally, the still elusive-but under intense research-association between biomarker levels, the special extent of brain lesions and the relevant clinical outcomes may further enhance prognosis in AIS and the ability to differentiate between ischemic stroke and stroke mimics [26].

## 6. Conclusions

Blood-based biomarkers hold promise as early detection tools for stroke diagnosis, and for predicting outcomes and informing clinical decision-making in IS. However, the current landscape underscores the need for improved methodological rigor and standardized reporting guidelines to ensure the validity and clinical applicability of prognostic biomarker studies. The development of biomarker panels and further exploration of associations with brain lesions and clinical outcomes represent crucial steps towards realizing the potential of the blood biomarkers in acute clinical setting, stroke triage and management. As research in this field continues to evolve, the integration of validated biomarkers into clinical practice aims at improving patient outcomes and reducing the burden of stroke-related morbidity and mortality. The ultimate goal is to develop a reliable, non-invasive diagnostic tool that can be easily implemented in several healthcare settings, thereby enhancing the overall efficiency and effectiveness of stroke management.

## Figures and Tables

**Figure 2 life-14-01247-f002:**
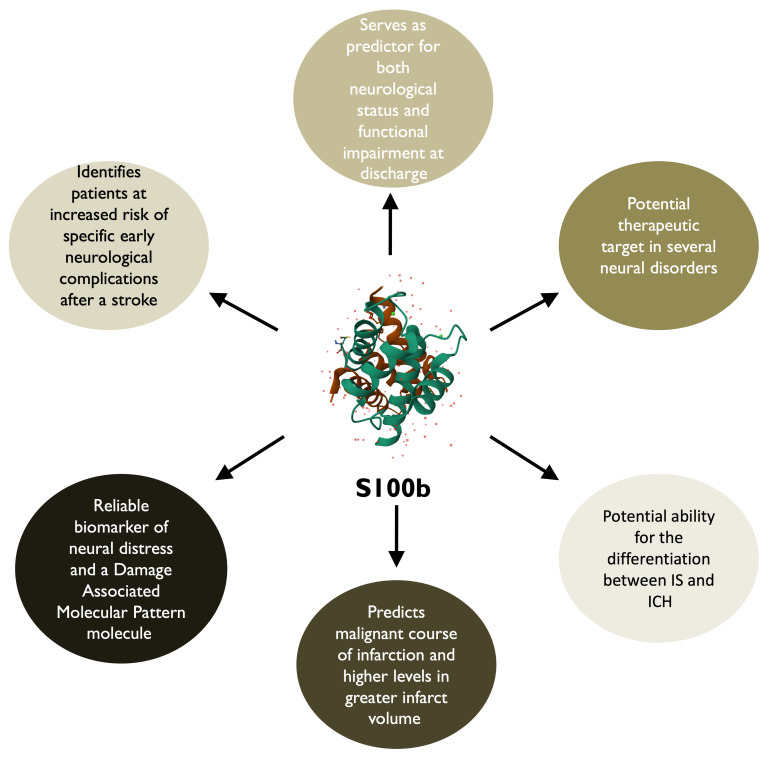
The role of S100B in patients with stroke [25,31,32,33,34,35].

**Table 3 life-14-01247-t003:** The criteria that different papers used to measure efficiency and efficacy of biomarkers in assessing patient progression after IS or ICH.

ASPECTS *, [53]
Barthel Index score **, [54]
CoRisk score ***, [55]
HINTS tests ^#^, [56]
modified Rankin score (mRS) ^##^, [57]
NIHSS ^###^, [58]
OSCP ^$^, [59]
Prehospital Stroke Score (PreSS), [60]
TOAST ^$$$^, [61]

NOTES: * Refers to Alberta Stroke Program Early CT Score (ASPECTS). To compute the ASPECTS score, one point is subtracted for any evidence of early ischemic change in any of ten designated brain regions. There are ten such brain regions: Caudate; Internal Capsule; Lentiform Nucleus; Insular Ribbon; Anterior MCA (i.e., middle cerebral artery supplied) cortex; MCA cortex lateral to the insular ribbon; Posterior MCA cortex; Anterior MCA territory immediately superior to Anterior MCA cortex, rostral to the basal ganglia; Lateral MCA territory immediately superior to MCA cortex lateral to the insular ribbon, rostral to the basal ganglia; Posterior MCA cortex territory immediately superior to posterior MCA cortex, rostral to the basal ganglia. ASPECTS Quantifies CT changes in early middle cerebral artery stroke. Although the ASPECTS score does not consistently predict treatment response or intracranial hemorrhage or offer nuanced prognostic information, patients with scores ≥8 have a better chance for an independent outcome. Patients with scores <8 (greater likelihood of poor functional outcome) may be helped in the early stages of care by transfer or therapy decisions. ** Refers to Barthel Index for Activities of Daily Living. The Barthel Index ranks the patient’s independence in the areas of Feeding, Bathing, Grooming, Dressing, Bowel control, Bladder control, Toilet use, Transfers (bed to chair and back), Mobility on level surfaces, Climbing stairs. *** Refers to copeptin-based parsimonious score to predict unfavorable outcome 3 months after an acute ischemic stroke. The score components are copeptin levels, age, NIH Stroke Scale, and recanalization therapy. Plasma Copeptin levels are measured within 24 h of acute ischemic stroke and before any recanalization therapy. In developing the CoRisk score the primary outcome of disability and death at 3 months was defined as modified Rankin Scale score of 3 to 6. # Refers to the outcome of the Head Impulse, Nystagmus and Test of Skew (HINTS) examination. The “HINTS examination is a useful tool in detecting acute, time-sensitive, central causes of vertigo, including posterior circulation strokes like lateral medullary syndrome. While most vertebrobasilar strokes are also accompanied by other signs (such as diplopia, dysarthria, dysphagia, motor, and sensory deficits) a proportion of cerebellar strokes present only with vertigo and subtle incoordination on examination. A positive HINTS exam has been reported to have a high sensitivity and specificity for the presence of a central cause of vertigo”. ## Refers to a scale for measuring the degree of disability or dependence in the daily activities of people who have suffered a stroke or other causes of neurological disability. It scores stroke from 0 (No symptoms) to 6 (Death). 1 means no significant disability despite symptoms whereby the patient is able to carry out all usual activities; 2 refers to slight disability whereby the patient is unable to carry previous activities but is able to look after their own affairs; 3 corresponds to moderate disability whereby the patient is able to walk without assistance albeit with some help; 4 is for moderately severe disability such as inability to walk without assistance and unable to independently attend own bodily needs; and 5 indicates severe disability whereby the patient is bedridden, incontinent and requiring continuous nursing care. ### Refers to NIH Stroke Scale. It is a scale that takes into account assessments of the Level of Consciousness (LOC) (assessment of responses to LOC Questions, and to LOC Commands); Best Gaze; of responses to Visual Instructions; of Facial Palsy; of Arm motor function; of Leg motor function; of Limb Ataxia; of Sensory functions; of best responses to Language Instructions; of Dysarthria; of Extinction and Inattention (formerly Neglect). The score can vary from 0 (completely normal) to 42 (most severe). $ OCSP stands for Oxford Community Stroke Project. It is a classification system used to categorize different types of cerebral infarctions based on presenting symptoms and signs. It assesses for lacunar infarcts (LACIs), total anterior circulation infarcts (TACIs) and partial anterior circulation infarcts (PACI), as well as posterior circulation infarcts (POCIs). The OSCP classification helps clinicians understand the specific type of stroke based on clinical features, which aids in diagnosis and treatment decisions. $$$ TOAST stands for the Acute Stroke Treatment classification system, and it provides valuable insights into ischemic stroke subtypes based on their etiology. There are five five main subtypes of stroke etiology that are identified by TOAST: 1. Large- (cerebral) Artery Atherosclerosis (LAA). 2. Cardioembolic Stroke (CE), often associated with conditions like atrial fibrillation or valvular heart disease. 3. Small-Vessel Disease (SVD)/Penetrating Artery Disease (PAD). SVD leads to lacunar infarcts, while PAD involves deeper brain structures. 4. Stroke of Other Determined Cause which includes strokes caused by specific factors such as vasculitis, dissection, or hypercoagulable states. 5. Stroke of Undetermined Cause (Cryptogenic Stroke), i.e., strokes whose exact etiology cannot be determined despite thorough evaluation. Diagnosis involves clinical assessment, brain imaging (CT/MRI), vascular imaging (CTA/MRA, neurosonology, DSA), cardiac imaging, and laboratory tests.

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
