# Peer review of "Current Trends in Stroke Biomarkers: The Prognostic Role of S100B and GFAP"

_life, 2024, doi:10.3390/life14101247_

Round 1

Reviewer 1 Report (Previous Reviewer 1)

Comments and Suggestions for Authors

The manuscript entitled “Current Trends in Stroke Biomarkers: The Prognostic Role of S100B and GFAP”, by Dr. Anogianakis and colleagues, has been substantially refined by the authors, by adding a general information on the role of S100B and GFAP. The review covers quite a little number of manuscripts, which is quite explained by a novelty of the field. I think, what the manuscript will be useful for a broad auditorium of clinicians and pre-clinicians.

Author Response

Thank you a lot for your comment. We really appreciate the feedback after the extensive major revision.

Reviewer 2 Report (Previous Reviewer 2)

Comments and Suggestions for Authors

This paper is a revised submission of a review of the role of S100B and GFAP as stroke biomarkers. The authors reviewed and summarised the recent (since 2000) clinical (strokes in humans) literature.  Some potential roles were identified.

There are a number of major issues the authors need to attend to:

1.      Abstract – to summarise the key findings more exactly (see 3. below)

2.      Line 105 - ‘recent’ should be added

3.      Lines 105-110 – I am still unclear what the aims of this paper are…lines 107-110 are best removed to avoid the distraction…

4.      Lines 137-146 – I don’t understand why these are placed here…. Lines 137-139 seem to be a rephrase of the aims, and thus should not be the ‘Methods’. Lines 139-144 seem more like a Conclusion and may also be best in the Abstract. As for lines 144-146…..

5.      Tables 1 and 2 – I (still) don’t understand how the references are sequenced – after [21] in the text, the table starts with 44, then 53…. I am glad to now see the studies are grouped (eg. IS, IS and ICH, and ‘-‘ which needs to be explained)  

6.      Text - ref [21] is followed by [29], then [23], etc etc – please re-do the ref numbering

7.      Discussion – needs to be more structured in a clinical relevant sequence with headings, based on Fig 1 and 2, tables 1 and 2. Eg distinguishing stroke from non-stroke; IS from ICH; stroke progression; etc etc

8.      If the paper is to help clinicians, the paper needs to be re-written for that audience

Author Response

This paper is a revised submission of a review of the role of S100B and GFAP as stroke biomarkers. The authors reviewed and summarised the recent (since 2000) clinical (strokes in humans) literature.  Some potential roles were identified.

There are a number of major issues the authors need to attend to:

  1. Abstract – to summarise the key findings more exactly (see 3. below)

-Thank you for your comment. The abstract was modified according to your recommendation.

  1. Line 105 - ‘recent’ should be added

-We thank you for your comment. The word “recent” was added to line 105.

  1. Lines 105-110 – I am still unclear what the aims of this paper are…lines 107-110 are best removed to avoid the distraction…

-We thank you for your comment. Lines 107-110 were removed to avoid the distraction as per your recommendation. The aim of the present review was modified into “The aim of the present review, therefore, is to analyze and summarize the recent relevant clinical literature on molecular markers of focal hypoxia with particular emphasis on the prognostic role of S100B protein and the GFAP in patients with stroke.”6

  1. Lines 137-146 – I don’t understand why these are placed here…. Lines 137-139 seem to be a rephrase of the aims, and thus should not be the ‘Methods’. Lines 139-144 seem more like a Conclusion and may also be best in the Abstract. As for lines 144-146.

-We would like to thank you for your comment. We deleted lines 137-139 as they seem to be a rephrase of the aims. In addition, lines 139-144 were transferred to the abstract, since they seem more like a conclusion, as you recommended.

  1. Tables 1 and 2 – I (still) don’t understand how the references are sequenced – after [21] in the text, the table starts with 44, then 53…. I am glad to now see the studies are grouped (eg. IS, IS and ICH, and ‘-‘ which needs to be explained)  

-Thank you for your comment. The tables are supposed to exist as an additional file in the manuscript/might be in the end of the text. The references in the text follow now a logical numbering sequence. If the study that is mentioned in the table is mentioned in the text, then it is numbered as in the text. There are also some studies that are not quite mentioned in the text and they are presented in the table. Those studies start from reference 54 if I am not mistaken.

  1. Text - ref [21] is followed by [29], then [23], etc etc – please re-do the ref numbering

-Thank you for your comment. We modified the numbering and 1 reference. 29 was changed to 19. Now, there is a logical numbering sequence in the references.

  1. Discussion – needs to be more structured in a clinical relevant sequence with headings, based on Fig 1 and 2, tables 1 and 2. Eg distinguishing stroke from non-stroke; IS from ICH; stroke progression; etc etc

-Thank you for your comment. Headings were added according to the content of the paragraphs.

-Regarding 5.2 (The role of s100b), the first paragraph discusses the proinflammatory properties of s100b and its role in distinguishing between IS and ICH. The second paragraph discusses its correlation with infarct volume. Third paragraph concerts its role in predicting early complications and functional outcomes.

-Regarding 5.3, the role of GFAP, the first paragraph highlights its role in prediction of hemorrhagic transformation and in distinguishing between hemorrhagic and ischemic stroke.

-The second paragraph demonstrates its prognostic role and its association with the extent of brain damage and neurological status

-I suppose, the paragraphs are now ordered according to the figures and in a clinical relevant sequence. We have also added subheadings according to your recommendation.

  1. If the paper is to help clinicians, the paper needs to be re-written for that audience

-Thank you for your comment. The paper was modified majorly according to your and the others reviewers’ comments.

Reviewer 3 Report (Previous Reviewer 5)

Comments and Suggestions for Authors

The authors have considerably improved the quality of this review.

Author Response

Thank you a lot for your comment. We really appreciate the feedback after the extensive major revision.

Round 2

Reviewer 2 Report (Previous Reviewer 2)

Comments and Suggestions for Authors

The authors have addressed my concerns

Author Response

Thank you for your comment.

This manuscript is a resubmission of an earlier submission. The following is a list of the peer review reports and author responses from that submission.

Round 1

Reviewer 1 Report

Comments and Suggestions for Authors

The review of Dr. Anogianakis and colleagues, entitled  “Current Trends in Stroke Biomarkers: The Role of S100B AND GFAP” describes recent literature data on potential stroke biomarkers, to differentiate between the types of stroke, and to estimate the risks of its further complications. The review scopes on a set of 41 research studies. The paper is definitely useful for clinical practice, since the two types of strokes share the third place among the leading causes of death.

The review is acceptable for “Life”. It will meet an interest of a broad readership. However, the paper will benefit greatly if the authors enlarge the Discussion by adding paragraphs on a normal role (i.e., in a healthy brain) for the discussed glial proteins, S100B and GFAP. Now, they are discussed as two enigmatic variables, which may be combined in a triage of stroke patients.

Minor points to improve are the following:

- Lines83 and 84 contain a strict words’ repetition, please re-phrase them.

- On Fig.4, it would be better to mark the name of the center protein (i.e. GFAP), similar to that on Fig. 2

Reviewer 2 Report

Comments and Suggestions for Authors

This paper is a review pf the role of S100B and GFAP as stroke biomarkers. The authors reviewed and summarised the recent (since 2000) clinical (strokes in humans) literature.  Some potential roles were identified.

There are a number of major issues the authors may need to attend to:

1.      Abstract – to summarise the key findings more exactly

2.      References - ref 1 is old, ref 2 is inappropriate – please recheck ALL references for relevance, accuracy and up-to-dateness

3.      Lines 92-93 – to emphasize it is the clinical (not basic science/animal) literature is being reviewed, and that it is of focal (and not global) hypoxia as in acute ischaemic stroke

4.      Tables 1 and 2 – I don’t understand how the references are sequenced – after [21] in the text, the table starts with 53, then 44, then 28…. Is there a logical plan? I suggest the studies be grouped (eg to distinguish stroke from non-stroke; IS from ICH, etc etc)  and be placed in a clinically relevant and logical sequence. So that it supports Figs 1 and 2

5.      Text - ref [27] is followed by [47], etc etc

6.      Discussion – needs to be more structured in a clinical relevant sequence, based on Fig 1 and 2, tables 1 and 2. Sensitivity and specificity etc need to be stated where possible

7.      Discussion of GFAP is very short (1 paragraph) compared to S100B

8.      There is inadequate discussion on these biomarkers add to the already existing clinical scales for the outcomes mentioned

9.      No discussion of limitations

10.  If the paper is to help clinicians, the paper needs to be re-written for that audience

Reviewer 3 Report

Comments and Suggestions for Authors

The manuscript provides a comprehensive review of the role of S100B and GFAP as biomarkers in stroke. However, it lacks a clear statement of how this review adds to the existing literature. Significant improvements are required. The authors should emphasize their review's unique contributions.

I have the following feedback for authors to consider:

Broader comments:

1. The manuscript covers a wide range of studies but does not critically evaluate the quality of the included studies. A more rigorous assessment of the methodological quality of the studies reviewed would strengthen the manuscript.

2. The process of data extraction and synthesis is not detailed. The authors should describe how data were extracted, how discrepancies were resolved, and the methods used for synthesizing the findings.

3. The process of data extraction and synthesis is not detailed. The authors should describe how data were extracted, how discrepancies were resolved, and the methods used for synthesizing the findings.

4. The manuscript discusses the potential clinical applications of S100B and GFAP but does not provide a clear description on the status and applicability to clinical practice. The authors should provide an overview of ongoing validation studies and the progress toward standardization.

Further/specific suggestions:

5. Abstract (Lines 14-27): Suggest including a brief mention of the search strategy and the number of studies reviewed to provide context.

6. Introduction (Lines 30-65): Suggesting giving a more detailed discussion of the current gaps in the literature and how this review aims to address them.

7. Methods (Lines 97-118): Suggets providing a detailed description of the data extraction process and the criteria used for assessing the quality of the included studies.

8. Results (Lines 119-126): Suggest including a summary table of the key findings from the reviewed studies, highlighting each study's strengths and limitations.

9. Discussion (Lines 236-348): Please include a critical evaluation of the methodological quality of the included studies and the implications for clinical practice.

10. Conclusion (Lines 349-360): The conclusion should provide specific recommendations for future research and clinical practice, based on the findings of the review.

11. Line 97: "A systematic electronic search..." should specify the exact search terms and Boolean operators used.

12. Line 109: "The eligibility of the retrieved studies..." should include a flow diagram of the study selection process.

13. Line 119: "As mentioned in the introduction..." should provide a summary of the key findings from the reviewed studies. 

14. Line 349:"Blood-based biomarkers hold promise..." should provide specific recommendations for future research and clinical practice.

Comments on the Quality of English Language

Moderate editing is required to improve readability and comprehension.

Reviewer 4 Report

Comments and Suggestions for Authors

Dear authors,

I would like to congratulate you for your manuscript!

Please allow me provide you a few suggestions.

1. Ethics and Data Availability Statements: The document should include explicit ethics and data availability statements to ensure transparency and reproducibility.

àRecommended Changes: Line to be added (suggested near the end of the Methods section or as a separate section):Ethics Statement: "This review was conducted in accordance with ethical guidelines and standards. No new data involving human or animal subjects were generated in this review.

  - Data Availability Statement: "All data discussed in this review are available from the cited sources. No new data were generated for this study."

2. Minor Language Edits

Issue: Minor language and typographical errors should be corrected to improve readability and professionalism.-->Recommended Changes:

- Line 45:

-Current: "Stroke survivors often experience permanent neurological damage, with significant impacts on their ability to work and overall quality of life."

  - Suggestion for revision: "Stroke survivors often experience permanent neurological damage, significantly impacting their ability to work and overall quality of life."

- Line 78:

  - Current: "These biomarkers are critical for predicting the onset, differentiating stroke subtypes (ischemic vs. hemorrhagic), and assessing clinical outcomes."

  - Suggestion for revision: "These biomarkers are critical for predicting the onset of stroke, differentiating stroke subtypes (ischemic vs. hemorrhagic), and assessing clinical outcomes."

 --Line 99, rephrase for clarity: "A systematic electronic search of the published research from January 2000 to February 2024 was conducted using the MEDLINE, Scopus, and Cochrane databases."

- Line 152:

  - Current: "Elevated S100B levels have been associated with worst outcomes and larger infarct volumes."

  - Suggestion for revision: "Elevated S100B levels have been associated with worse outcomes and larger infarct volumes."

3. Figures/Tables/Images Line 220 (Table 1):

  - Current: "Table 1: Biomarkers in Stroke Diagnosis."

  - Suggestion for revision: "Table 1: Overview of Biomarkers in Stroke Diagnosis and Prognosis."

- Line 255 (Figure 2):

  - Current: "Figure 2: Levels of S100B in Stroke Patients."

  - Suggestion for revision: "Figure 2: Comparative Levels of S100B in Ischemic and Hemorrhagic Stroke Patients."

4. Title: Consider specifying that the review focuses on the prognostic value of S100B and GFAP in acute ischemic stroke (AIS).

5. Introduction: Lines 1-10: Improve clarity and flow. For example, rephrase to: "Biological markers have been extensively studied since 1977 when the term 'biological marker' first appeared in Medline. However, most molecular markers of hypoxia related to stroke are neuronal markers for degenerative diseases of the nervous system rather than stroke-specific markers."

6.Methods:Line 109: Correct the punctuation and improve sentence structure. For example, "The eligibility of the retrieved studies was independently assessed by two investigators (N.T, K.B) according to prespecified criteria. The most relevant full-text articles investigating the prognostic significance of S100B and GFAP (measured either as dichotomous or continuous variables) in patients with ischemic stroke, irrespective of the stroke type, were included in this review."

 7. Results:Line 121: Correct punctuation and ensure consistency. For example, "As mentioned in the introduction, the aim of the present review is to analyze and summarize the relevant literature on molecular markers of hypoxia with particular emphasis on the S100B protein (Table 1) and GFAP (Table 2)."

8. Discussion:Lines 187-214: Clarify the role and findings related to GFAP. For example, "GFAP is a structural protein in mature astrocytes in the central nervous system, characterized by a filament length of approximately 8-9 nm. The gene encoding GFAP is located on chromosome 17q21. In normal astrocytes, GFAP is expressed as a non-soluble monomeric protein comprising 432 amino acids with a molecular weight of 49.8-53 kDa."

9. Conclusion:Lines 215-218: Ensure a strong and clear conclusion summarizing the importance of S100B and GFAP in prognosticating AIS outcomes. For example, "In conclusion, both GFAP and S100B are promising biomarkers for predicting outcomes in acute ischemic stroke. Combining NIHSS at 24 hours with GFAP at 48 hours yields the best prediction for negative outcomes."

Reviewer 5 Report

Comments and Suggestions for Authors

Thank you for carrying out such an interesting work, however, I have some doubts before proceeding with your publication.

Inclusion and exclusion criteria should be correctly indicated.

Add a flow chart showing the process of searching and selecting articles.

Acronyms are indicated in the text itself, not at the bottom of the page.

Table 1 is incomplete. The methodology used for each article should be indicated in the table.

Have these selected articles been selected or screened using any scale to ensure their quality?